# Antifungal Activity of Soft Tissue Extract from the Garden Snail *Helix aspersa* (Gastropoda, Mollusca)

**DOI:** 10.3390/molecules27103170

**Published:** 2022-05-16

**Authors:** Hoda H. Abd-El Azeem, Gamalat Y. Osman, Hesham R. El-Seedi, Ahmed M. Fallatah, Shaden A. M. Khalifa, Mohamed M. Gharib

**Affiliations:** 1Department of Zoology, Faculty of Sciences, Menoufia University, Shebin El-Kom 32512, Egypt; dr.gyosman@yahoo.com; 2Pharmacognosy Group, Department of Pharmaceutical Biosciences, Biomedical Centre, Uppsala University, P.O. Box 591, SE 751 24 Uppsala, Sweden; 3International Research Center for Food Nutrition and Safety, Jiangsu University, Zhenjiang 212013, China; 4International Joint Research Laboratory of Intelligent Agriculture and Agri-Products Processing, Jiangsu Education Department, Jiangsu University, Zhenjiang 212013, China; 5Department of Chemistry, Faculty of Science, Menoufia University, Shebin El-Kom 32512, Egypt; 6Department of Chemistry, College of Science, Taif University, P.O. Box 11099, Taif 21944, Saudi Arabia; a.fallatah@tu.edu.sa; 7Department of Molecular Biosciences, The Wenner-Gren Institute, Stockholm University, SE 106 91 Stockholm, Sweden; shaden.khalifa.2014@gmail.com; 8Department of Botany, Faculty of Sciences, Menoufia University, Shebin El-Kom 32512, Egypt; gharieb2000@yahoo.com

**Keywords:** Helix aspersa, soft tissue, extract, Candida albicans, Aspergillus flavus, Aspergillus brasiliensis

## Abstract

Gastropods comprise approximately 80% of molluscans, of which land snails are used variably as food and traditional medicines due to their high protein content. Moreover, different components from land snails exhibit antimicrobial activities. In this study, we evaluated the antifungal activity of soft tissue extracts from *Helix aspersa* against *Candida albicans*, *Aspergillus flavus*, and *Aspergillus brasiliensis* by identifying extract components using liquid chromatography-tandem mass spectrometry (LC-MS-MS). Two concentrations of three extracts (methanol, acetone, and acetic acid) showed antifungal activity. Both acetone (1 g/3 mL) and acetic acid extracts (1 g/mL) significantly inhibited *C.*
*albicans* growth (*p* = 0.0001, 5.2 ± 0.2 mm and *p* = 0.02, 69.7 ± 0.6 mm, respectively)*. A. flavus* and *A. brasiliensis* growth were inhibited by all extracts at 1 g/mL, while inhibition was observed for acetic acid extracts against *A. brasiliensis* (*p* = 0.02, 50.3 ± 3.5 mm). The highest growth inhibition was observed for *A. flavus* using acetic acid and acetone extracts (inhibition zones = 38 ± 1.7 mm and 3.1 ± 0.7 mm, respectively). LC-MS-MS studies on methanol and acetone extracts identified 11-α-acetoxyprogesterone with a parent mass of 372.50800 *m*/*z* and 287.43500 *m*/*z* for luteolin. Methanol extracts contained hesperidin with a parent mass of 611.25400 *m*/*z*, whereas linoleic acid and genistein (parent mass = 280.4 and 271.48900 *m*/*z*, respectively) were the main metabolites.

## 1. Introduction

For hundreds of years, land snails have been used as a food source and for medical treatments. Land snails are pharmacologically and medicinally important [1] as they are high in protein, which they use to combat different environmental conditions [2]. Previously, seven crude proteins were extracted from different snails. The most active crude proteins identified in the land snail, *Cryptozona bistrialis*, were active against different bacterial and fungal pathogens [3]. Therefore, snails should be considered important bioactive compound sources with safe pharmaceutical applications as polypeptides, proteins, and glycans from snail mucus could function as promising candidates for some dermal diseases [4].

Similarly, protein and peptide components from the hemolymph and mucus of garden snails showed antimicrobial activity [5]. Slime from the African giant land snail *Archachatina marginata* demonstrated antibacterial activity against *Escherichia coli* (inhibition zone = 15.2 mm), *Klebsiella* sp. (inhibition zone = 14.2 mm), *Pseudomonas aeruginosa* (13.0 mm), and *Proteus mirabilis* (13.3 mm) [6].

Three protein fractions from the marine snail *Rapana venosa* were effective against *Aspergillus niger*, *Botrytis cinerea*, and *Candida albicans* [7].

Two protein fractions from the mucus of *Helix aspersa* (>20 kDa) and a peptide fraction from the hemolymph of *Helix lucorum* (<10 kDa) were isolated, and their antibacterial activities against *E.*
*coli* and *Brevindomonas diminita* were characterized. Their minimum inhibitory concentration values ranged from 145 to 682.5 µg/mL for *E. coli* and *B*. *diminuta* [8]. In hemolymph from *H. lucorum*, nuclear magnetic resonance metabolic analysis and tandem mass spectrometry (MS-MS) were used to detect metabolites (<1 kDa and <3 kDa) with antioxidant and antimicrobial activities [9]. Ref. [10] isolated a 1485.26 Da peptide (Cm-p1, sequence = SRSELIVHQR) from a crude extract of the marine snail, *Cenchritis muricatus*, which demonstrated antifungal activity against filamentous fungi and yeast. Moreover, the antifungal activity of a crude methanol extract from *Cypraea* spp. Against *C. albicans* and *A. niger* was similarly demonstrated [11]. In other work, ethanol, acetone, and methanol crude extracts from *Babylonia spirata* were effective against *A. flavus* and *C. albicans* [12].

Fungal infections may occur in the hair, nails, and skin and may cause serious diseases [13]. The total number of individuals who experience different fungal infections is approximately 1,000,000,000. Fungal diseases may kill more than 1.5 million and affect over a billion people. The health consequences of serious fungal infections include asthma, serious chronic illness, blindness and corticosteroid therapies. The modern global estimates have found ~700,000 cases of invasive candidiasis and 3,000,000 cases of chronic pulmonary aspergillosis [14]. *C.*
*albicans* causes superficial, deep tissue, and invasive candidiasis [15], and its mortality rate is 35–60% due to disseminated candidiasis [16]. Fungal diseases such as aspergillosis cause high mortality rates due to respiratory disorders [17]. Bronchopulmonary aspergillosis caused by *A. fumigatus*, *A. flavus*, and *A. niger* colonies in the lung mucosa spread infection, with a mortality rate of 50–90% [18]. Therefore, antifungal properties from bioactive compounds in invertebrates could provide new directions for medical treatments and scientific research.

A novel neurotoxin (conotoxin TVIIA) was extracted from the sea snail *Conus tulipa* [19] and was identified as belonging to the conotoxin family, which is composed of six-cysteine/four-loop structures with pharmacological activities. The molecules block ionic calcium, sodium, and potassium channels [20,21]. The contryphan-Vn peptide was extracted from the Mediterranean snail, *Conus ventricosus*; it contained a D-tryptophan residue and maintained its five-residue intercystine loop. The peptide was bound to potassium channels and displayed distinct molecular targets [22].

In this study, we isolated soft tissue extracts from *H.*
*aspersa* and identified antifungal activities against *C. albicans*, *A.*
*flavus*, and *A.*
*brasiliensis.* Extract components were identified using liquid chromatography-mass spectrometry (LC-MS-MS). 

## 2. Results

Two concentrations, 1 g/mL and 1 g/3 mL, of three crude extracts (methanol, acetone, and acetic acid extracts) from the soft body (viscera) of *H. aspersa* were used to test for antifungal effects against *C. albicans*, *A. flavus*, and *A. brasiliensis*.

Individual solvents exerted no effects on fungal growth of *C. albicans* except for acetic acid (inhibition zone = 26 ± 0.3 mm). All extracts showed antifungal activity when compared with the controls (Figure 1). Methanol and acetone extracts caused insignificant *C. albicans* growth inhibition (inhibition zone = 20.06 ± 0.1 mm and 59.4 ± 1.4 mm at 1 g/mL). Acetone extracts showed significant inhibition zones against C. *albicans* (*p* = 0.0001, 5.2 ± 0.2 mm at 1 g/3 mL). In a concentration-dependent manner, acetic acid generated significant inhibition zones against *C. albicans* (ANOVA, p = 0.02, 69.7 ± 0.6 mm at 1 g/mL (1:1)) (Figure 2).

The antifungal drug fluconazole (25 µg/mL) inhibited *C. albicans* growth at 19 ± 0.1 mm (Figure 1). 

The antifungal drug fluconazole (25 µg/mL) inhibited *A. brasiliensis* and *A. flavus* growth at 17 ± 0.4 mm and 14 ± 0.1 mm, respectively. The solvents had no effects on fungal growth except for acetic acid (inhibition zone = 15 ± 0.1 mm and 10 ± 0.5 mm for *A. brasiliensis* and *A. flavus*, respectively) (Figure 1). Both *A. flavus* and *A*. *brasiliensis* growth was inhibited by all extracts at 1 g/mL. Still, inhibition was significant for the acetic acid extract against *A. brasiliensis* (ANOVA, *p* = 0.02, 50.3 ± 3.5 mm) (Figure 1 and Figure 3).

The most significant growth inhibition was observed for *A. flavus* against acetic acid and acetone extracts (inhibition zones = 38 ± 1.7 mm and 3.1 ± 0.7 mm, respectively).

### Identified Compounds Using LC-MS-MS Analysis

Plant metabolome mass profiles were screened using the Global Natural Products Social Molecular Networking (GNPS) database (Appendix A, Table 1). Five compounds were identified: genistein, luteolin, and hesperidin are flavonoid compounds, linoleic acid is a fatty acid, and 11-α-acetoxyprogesterone is a steroid compound.

## 3. Discussion

Our data indicated that all extracts from the snail *Helix aspersa* have bioactive ingredients which exerted antifungal activities against *C.*
*albicans*, *A.*
*flavus*, and *A.*
*brasiliensis* in a concentration-dependent manner. Furthermore, our results were supported by [12], who extracted bioactive compounds from the marine snail, *B.*
*spirata*. Moreover, the antibacterial activity of crustacean and molluscan methanol extracts was higher when compared with water extracts [3]. 

The antifungal activity of extracts may have been related to their metabolite components, as demonstrated by LC-MS-MS; the most common component was hesperidin, with a parent mass = 611.25400. This flavanone glycoside induced apoptosis and cell cycle arrest [27,28] and showed antifungal, antiviral, antihelminthic, antioxidant, and molluscicidal activities toward *Schistosoma mansoni*. Genistein is also present in plants and humans and is an isoflavone with antihelminthic qualities [29]. Genistein was speculated to inhibit fungal growth due to its apoptosis-inducing characteristics [30]. Genistein also induced apoptosis in human cancer cells by triggering both caspase-3 and caspase-9 activity, causing cell death by inhibiting NF-κB signaling and altering levels of the antiapoptotic protein Bcl-2 and proapoptotic protein. In addition, genistein excited oxidative stress-induced apoptosis by increasing nitric oxide production and its bioavailability [31,32,33]. Furthermore, genistein is an estrogenic isoflavone found in the molluscan tissues, and its receptors were isolated from the cerebral ganglia of the gastropod *Thais clavigera* [34]. 

The mass of 11-α-acetoxyprogesterone was 372.50800. Progesterone is a steroid hormone that functions as a substantial metabolic intermediate during corticosteroid and sex hormone production and also functions as a neurosteroid [35,36]. The antifungal effects of this compound are facilitated by the presence of high-affinity progesterone-binding sites in the plasma membrane, as confirmed by [37] in *Rhizopus nigricans*. A corticosteroid-binding protein and steroid receptor were identified by [38] in *C. albicans* [39] as well as other fungal species. Progesterone is considered an effective fungal growth inhibitor [40]. Its growth inhibition is related to reduced intracellular cyclic adenosine monophosphate(cAMP) levels that are crucial arbiters of fungal growth and responses to nutritional stress [41].

Luteolin is a tetrahydroxyflavone and exerts antifungal effects against *C.*
*albicans* [42]. The fungal growth inhibition was reported whereby plasma membrane disruption led to membrane permeability changes and excess reactive oxygen species (ROS) production [43]. The ROS effects are mainly directed toward membrane lipids in *C. albicans*, which generate lipid hydroperoxides and thus lipid peroxidation [44]. The induction of mitochondrial dysfunction via inhibited mitochondrial electron transport chain reduces ATP synthesis and causes cell death by inhibiting cell wall formation, cell division, and RNA and protein synthesis [45]. 

Linoleic acid is a polyunsaturated essential fatty acid that reduces biomass production in *Rhizoctonia solani*, *Pythium ultimum*, *Pyrenophora avenae*, and *Crinipellis perniciosa*, while 1000 µM linoleic acid reduces mycelial growth in *R. solani*, *P. ultimum*, and *P. avenae* [46]. Linoleic acid is a structural component related to membrane fluidity functions and directly interacts with fungal cell membranes by entering lipid bilayers, increasing membrane fluidity, disorganizing cell membranes, and causing cell disintegration [47]. Antifungal fatty acids can replace synthetic agrochemicals that control fungal pathogens [48]. The antifungal activity of linoleic acid was reported against *Aspergillus amylovorus* (NRRL 5813) and *A. flavus* (NRRL 3518) [49,50]) reported its antifungal activity against *C.*
*albicans* and *Candida*
*parapsilosis* [50].

In conclusion, soft tissue extract metabolites from *H.*
*aspersa* are promising antifungal agents.

## 4. Materials and Methods

### 4.1. Sample Collection and Care

Wild *H. aspersa* were collected from infested ornamental fruit and grass plants in Menoufia governorate, Egypt. Snails were maintained under the following laboratory conditions: 12 h:12 h light/dark photoperiod, room temperature, and relative humidity, 85%. Approximately 10–15 individuals (average weight = 4 g) were placed in glass containers (15 × 15 × 22 cm) filled with moist soil (sandy loam) and covered in muslin for ventilation. Fresh lettuce leaves (*Lactuca sativa*) and water for soil humidity were supplied daily. Waste food and fecal matter were removed at the end of every other day. Snails were acclimatized under laboratory conditions for at least 4 weeks. 

### 4.2. Tissue Extraction

Snails were rinsed, and the shells removed. The soft body (viscera) was cut into small pieces and homogenized in different solvents, methanol, acetone, and acetic acid [3,12]. Solvent tissue homogenization was performed on ice. Two concentrations of the three solvents (methanol, acetone and acetic acid) were used: 1 and 3 mL for one gram of tissue (viscera). Homogenates were centrifuged at maximum speed in a refrigerated centrifuge, added to clean tubes, and maintained on ice till the inculcation of samples. Crude extracts were used for antifungal assays against the fungal pathogens, *C.*
*albicans*, *A.*
*flavus*, and *A. brasiliensis*.

### 4.3. Antifungal Assays

Two concentrations (1 g/mL and 1 g/3 mL) of the three crude extracts (methanol, acetone and acetic acid extracts) were prepared to test the growth inhibition of *C.*
*albicans*, *A. flavus*, and *A. brasiliensis* (ATCC 16404) clinical isolates.

Antifungal activity was determined using the agar well diffusion method [51]. Nutrient agar plates were aseptically spread with 24 h cultures from respective pathogens and incubated for 15 min in a laminar chamber to facilitate absorption. Next, 5 mm wells were aseptically cut in the agar for visceral extract addition—100 µL of each extract was added, and plates were left for 1 h for infusion. Then, plates were incubated at 37 °C for 24 h, after which extract inhibition zone diameters were measured in mm. Fluconazole (25 µg/mL) was used as a positive control. All tests were performed in triplicate, and mean values were recorded for statistical analysis.

### 4.4. Chemical Analysis of Extracts

Methanol and acetone extracts were analyzed using LC-MS-MS. A Shimadzu LC-10 high-performance liquid chromatography instrument with a Grace Vydac Everest Narrowbore C18 column (100 mm × 2.1 mm i.d., 5 µm, 300 Å) was connected to an LCQ electrospray ion trap MS (Thermo Finnigan, San Jose, CA, USA). Raw data files were converted to mzXML format using MSConvert from the ProteoWizard suite (http://proteowizard.sourceforge.net/tools.shtml, (accessed on 1 March 2022).). A molecular network was created using the GNPS online workflow. Spectra in the network were then searched against GNPS spectral libraries and published data [52].

### 4.5. Statistical Analysis

Data were expressed as the mean ± standard deviation and analyzed using Statgraphics Centurion XVI (Stat-Point Technologies Inc., Warrenton, VA, USA). Statistical analysis was conducted using a two-way analysis of variance to identify differences between tissue extracts and solvents and tissue extracts and concentrations. A probability *p* ≤ 0.05 level was accepted as significant.

## Figures and Tables

**Figure 1 molecules-27-03170-f001:**
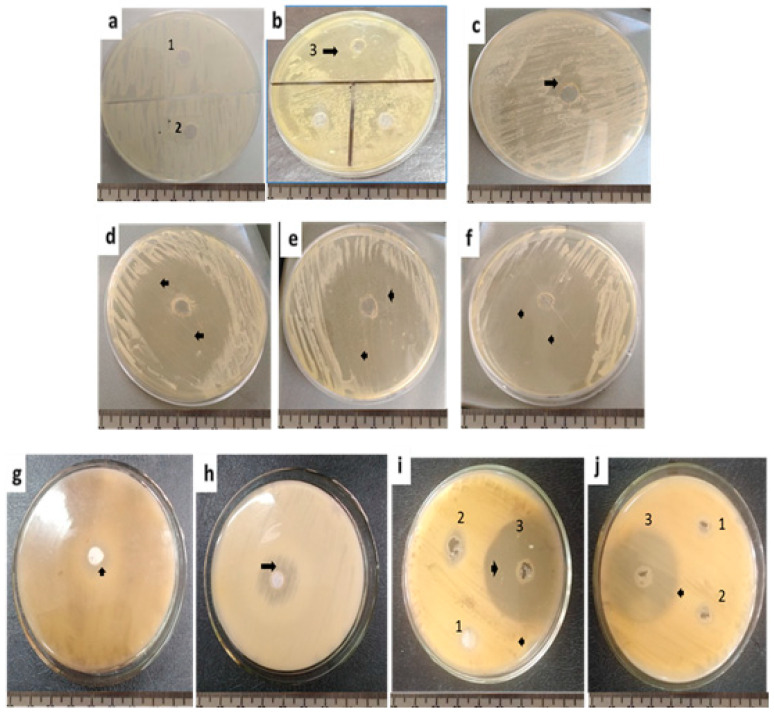
Antifungal activity of the methanol, acetone and acetic acid viscera extracts of the *H. aspersa*. (**a**) *C. albicans* with methanol (1), acetone (2) and (**b**) acetic acid (3). (**c**) *C. albicans* growth with the antifungal drug, namely fluconazole (25 µg/mL). (**d**) *C. albicans* growth with methanolic viscera extract. (**e**) *C. albicans* growth with acetone viscera extract. (**f**) *C. albicans* growth with acetic acid viscera extract. (**g**) *A. brasiliensis* growth with the antifungal drug fluconazole (25 µg/mL). (**h**) *A. flavus* growth with the antifungal drug fluconazole (25 µg/mL). (**i**) *A. brasiliensis* growth with methanol viscera extract (1), acetone viscera extract (2) and acetic acid viscera extract (3). (**j**) *A. flavus* growth with methanol viscera extract (1), acetone viscera extract (2) and acetic acid viscera extract (3). Arrowheads are pointing to the inhibition zone.

**Figure 2 molecules-27-03170-f002:**
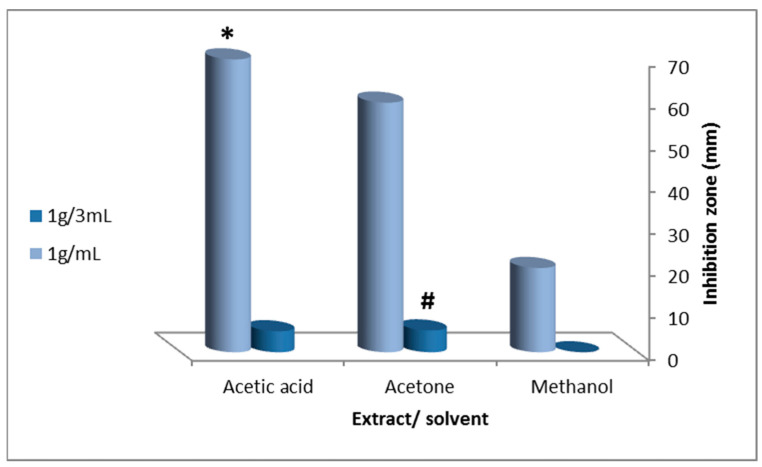
Antifungal activity of viscera (two concentrations) extracted by (methanol, acetone and acetic acid) of the garden snail, *H**. aspersa*, against *Candida albicans*. The values are expressed asmean ± SD; 
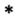
 indicates a significant difference between tissue extract and concentration; 
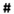
 indicates a significant difference between extracting solvents (ANOVA, *p* ≤ 0.05).

**Figure 3 molecules-27-03170-f003:**
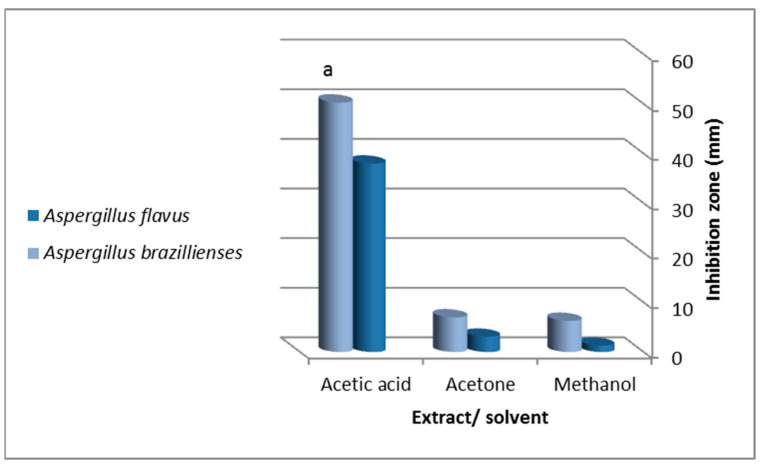
Antifungal activity of viscera extracted by (methanol, acetone and acetic acid) of the garden snail, *Helix aspersa* against *Aspergillus flavus* and *Aspergillus brasiliensis*. Values are expressed as mean ± SD; “a” indicates a significant difference between extracting solvents (ANOVA, *p* = 0.02).

**Table 1 molecules-27-03170-t001:** The identified metabolites’ parent masses and fragments from the raw mass spectrum compared with that of the molecular networking database and data published.

Compound Name	Parent Mass (*m*/*z*)	M.F	Fragments from the Raw Mass Spectrum	Fragments from GNPS Database	Reference
Genistein ^1,2^	271.48900	C_15_H_10_O_5_	144.96, 148.98, 152.92, 214.96, 255.03, 242.95	144.98, 148.39, 152.94, 215.06, 255.01, 242.02	
Luteolin ^1,2,3,4^	287.43500	C_15_H_10_O_6_	110.98, 134.96, 152.90, 161.00, 241.00, 259.00, 269.00	111.00, 135.00, 153.00, 160.98, 241.04, 259.04, 269.05	[23,24]
Linoleic acid ^1,2^	280.4 g	C_18_H_32_O_2_	153,95, 176.90, 219.07, 286.03	153,02, 177.06, 219.07, 286.10	
11-alpha-Acetoxyprogesterone ^1,2,3,4^	372.50800	C_23_H_32_O_4_	172.95, 277.14, 294.19, 312.15	173.10, 277.18, 295.20, 313.22	[25]
Hesperidin	611.25400	C_28_H_34_O_15_	303.05, 345.05, 413.06, 449.07, 465.07, 489.10, 557.20, 593.29	303.05, 345.05, 413.06, 449.07, 465.07, 489.10, 557.20, 593.29	[26]

^1^ sample codex “methanol extract (1 g/3 mL)”, ^2^ “methanol extract(1 g/mL)”, ^3^ “acetone extract (1 g/3 mL)”, ^4^ “acetone extract (1 g/mL)”.

## Data Availability

Not applicable.

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
