# Peer review of "Antifungal Activity of Soft Tissue Extract from the Garden Snail *Helix aspersa* (Gastropoda, Mollusca)"

_molecules, 2022, doi:10.3390/molecules27103170_

Round 1
Reviewer 1 Report
The authors performed the assays including a control for the solvents (acetic acid, methanol, and acetone) used to extract the metabolites from Helix aspersa. However, acetic acid has an antimicrobial effect on Candida albicans, Aspergillus flavus, and Aspergillus brasiliensis (lines 103-105 and 118-121). This should be discussed in the discussion section and mentioned in the material and methods section. The acetic acid, methanol, and acetone concentrations applied should be mentioned in both sections.
The legend in figure 3 is not perceptible. Controls with the solvents, acetic acid, methanol, and acetone for Candida albicans, Aspergillus flavus, and Aspergillus brasiliensis were performed in which plates? The legend should be improved.
In the results section line 100-102 the concentration of the 3 crude extracts must be mentioned.
In fig 1 the concentrations are expressed like this: 1gm/3ml and 1gm/ml the authors meant 1mg/3ml and 1gm/ml? Please revise the manuscript regarding the units written for all the concentrations, this must be corrected.
Author Response
Reviewer1:
- The authors performed the assays including a control for the solvents (acetic acid, methanol, and acetone) used to extract the metabolites from Helix aspersa. However, acetic acid has an antimicrobial effect on Candida albicans, Aspergillus flavus, and Aspergillus brasiliensis(lines 103-105 and 118-121). This should be discussed in the discussion section and mentioned in the material and methods section. The acetic acid, methanol, and acetone concentrations applied should be mentioned in both sections.
Response:-
Lines 217-218: “Two concentrations of the three solvent (methanol, acetone and acetic acid) were used: 1 and 3mL for one gram of tissue (viscera)”
Lines 223-225: “Two concentrations (1g/mL and 1g/3mL) of the three crude extracts (methanol, acetone and acetic acid extracts) were prepared to test the growth inhibition of C. albicans, A. flavus, and A. brasiliensis (ATCC 16404) clinical isolates.”
Concerning the discussion section we explained that all solvents exerted ingredients from the tissue and discussed the activity of each ingredient, not the effect of the solvents themselves. More over the antimicrobial activity of (acetic acid +tissue) was described
This applies to A. brasiliensis and A. flavus,
- The legend in figure 3 is not perceptible. Controls with the solvents, acetic acid, methanol, and acetone for Candida albicans, Aspergillus flavus, and Aspergillus brasiliensiswere performed in which plates? The legend should be improved.
Response: The legend is adjusted
(a) C. albicans with methanol (1), acetone (2) ( b) acetic acid (3).
To no repeat the pictures , The control solvents has no effect on Asperagillus flavus, and Asperagillus brasiliensis except acetic acid so it was mentioned in the results section as a number.
- In the results section line 100-102 the concentration of the 3 crude extracts must be mentioned.
Response:- Adjusted
Lines 100-101: “Two concentrations, 1g/mL and 1g/3mL of three crude extracts (methanol, acetone, and acetic acid extracts)”
- In fig 1 the concentrations are expressed like this: 1gm/3ml and 1gm/ml the authors meant 1mg/3ml and 1gm/ml? Please revise the manuscript regarding the units written for all the concentrations, this must be corrected.
Response:-We meant 1g/mL and 1g/3mL
Reviewer 2 Report
Some minor errors need to be improved. For example, the 'ml' should be changed to 'mL' throughout the MS.
Author Response
Reviewer 2:
Some minor errors need to be improved. For example, the 'ml' should be changed to 'mL' throughout the MS.:
Response: Adjusted
Round 2
Reviewer 1 Report
Some minor points must be revised in the manuscript:
Please correct the units in Table 1: sample codex “methanol extract (1gm/mL) ”, 2: “methanol extract(1gm/1mL)”, 3: “acetone extract 150 (1gm/3mL)”, 4: “acetone extract (1gm/mL)”
Line 119-120-Please refer to the figure -"Solvents had no effects on fungal growth except for acetic acid (inhibition zone = 15 ± 0.1 mm and 10 ± 0.5 mm for A. brasiliensis and A. flavus, respectively)".
Author Response
Reviewer1:
Some minor points must be revised in the manuscript:
- Please correct the units in Table 1:sample codex “methanol extract (1gm/mL) ”, 2: “methanol extract(1gm/1mL)”, 3: “acetone extract 150 (1gm/3mL)”, 4: “acetone extract (1gm/mL)”
Response: adjusted
Lines 151-152: “1: sample codex “methanol extract (1g/3mL) ”, 2: “methanol extract(1g/mL)”, 3: “acetone extract (1g/3mL)”, 4: “acetone extract (1g/mL)””
- Line 119-120-Please refer to the figure -"Solvents had no effects on fungal growth except for acetic acid (inhibition zone = 15 ± 0.1 mm and 10 ± 0.5 mm for A. brasiliensis and A. flavus, respectively)".
Response: Done
This manuscript is a resubmission of an earlier submission. The following is a list of the peer review reports and author responses from that submission.
Round 1
Reviewer 1 Report
The manuscript has some scientific faults that have to be corrected; and the assays must be repeated. The authors should perform assays including a control for the solvents used to extract the metabolites from Helix aspersa. The solvents acetic acid, methanol, and acetone have an antimicrobial effect on several microorganisms and the authors didn´t test the antimicrobial activity of these solvents on their target fungi, i.e., Candida albicans, Aspergillus flavus, and Aspergillus brasiliensis.
The authors should have performed the disk diffusion assay with each solvent in the concentrations presents in the extracts of the snail Helix aspersa soft tissue to ensure that it does not have any antifungal activity and that whatever antifungal activities will be shown in the different assays can be attributed to the compounds in extracts of Helix aspersa itself. Therefore, the authors cannot assume that the antifungal activity is only related to the metabolites present in the Helix aspersa soft tissue.
Considering this, I suggest that the manuscript in its present form should be rejected. Nevertheless, if the authors perform the controls suggested and the results show that the concentration of the solvents present in the extracts of the snail Helix aspersa soft tissue is not the reason for the antifungal activity observed, the article can be again revised.
Reviewer 2 Report
This study is an original and important contribution. However, it cannot be accepted in its current state. The authors must carry out a deep revision of the English language and improve the presentation of all sections of the manuscript. Authors are strongly encouraged to have their manuscript corrected by a native English speaker.
Reviewer 3 Report
1.There are serious defects in the experimental design of this paper. The research object selected in the manuscript is snail viscera/(solvend), to detect the antifungal activity of the extracts (methanol, acetone and acetic acid extracts/solvents) , as we known,the feed of snails is plants, The experiment should avoid the antibacterial effect of snail food in this manuscript. Although the author fed lettuce leaves for 4 weeks, it can not be ruled out that the antibacterial substances in the food have an effect on the two strains of bacteria in the manuscript.
2.Lettuce leaves are also introduced in the experiment as the food, which is lack of blank control experiment
3.There is also a lack of soil blank control experiment